# Recent Trends in Polymeric Foams and Porous Structures for Electromagnetic Interference Shielding Applications

**DOI:** 10.3390/polym16020195

**Published:** 2024-01-09

**Authors:** Marcelo Antunes

**Affiliations:** Department of Materials Science and Engineering, Poly2 Group, Technical University of Catalonia (UPC BarcelonaTech), ESEIAAT, C/Colom 11, 08222 Terrassa, Spain; marcelo.antunes@upc.edu; Tel.: +34-93-7398056

**Keywords:** polymeric foams, porous structures, microcellular foams, 3D printing, carbon-based nanofillers, nanohybrids, EMI shielding, EM wave absorption

## Abstract

Polymer-based (nano)composite foams containing conductive (nano)fillers limit electromagnetic interference (EMI) pollution, and have been shown to act as good shielding materials in electronic devices. However, due to their high (micro)structural complexity, there is still a great deal to learn about the shielding mechanisms in these materials; understanding this is necessary to study the relationship between the properties of the microstructure and the porous structure, especially their EMI shielding efficiency (EMI SE). Targeting and controlling the electrical conductivity through a controlled distribution of conductive nanofillers are two of the main objectives when combining foaming with the addition of nanofillers; to achieve this, both single or combined nanofillers (nanohybrids) are used (as there is a direct relationship between electrical conductivity and EMI SE), as are the main shielding mechanisms working on the foams (which are expected to be absorption-dominated). The present review considers the most significant developments over the last three years concerning polymer-based foams containing conductive nanofillers, especially carbon-based nanofillers, as well as other porous structures created using new technologies such as 3D printing for EMI shielding applications. It starts by detailing the microcellular foaming strategy, which develops polymer foams with enhanced EMI shielding, and it particularly focuses on technologies using supercritical CO_2_ (sCO_2_). It also notes the use of polymer foams as templates to prepare carbon foams with high EMI shielding performances for high temperature applications, as well as a recent strategy which combines different functional (nano)fillers to create nanohybrids. This review also explains the control and selective distribution of the nanofillers, which favor an effective conductive network formation, which thus promotes the enhancement of the EMI SE. The recent use of computational approaches to tailor the EMI shielding properties are given, as are new possibilities for creating components with varied porous structures using the abovementioned materials and 3D printing. Finally, future perspectives are discussed.

## 1. Introduction

Among typically used materials for EMI shielding, carbon-based materials and metals are two of the most widely used, owing to their intrinsically high electrical conductivity and high electromagnetic (EM) wave reflection [1,2]. Being non-conductive, common polymers are generally non-effective or less effective, unless when combined with other functional materials and/or components. Although the non-conductive issue can be partially solved by using intrinsically electrically conductive polymers, such as polythiophene, polyanilines, or poly(p-phenylene vinylene), these polymers display limited mechanical and thermal performances, considerably reducing their range of applications. Owing to developments made in the last 20–25 years, concerning different types of micro- and especially nano-sized carbon-based fillers, such as carbon nanotubes and more recently graphene-based materials (graphene oxide, reduced graphene oxide, graphene nanoplatelets, etc.), their combination with non-conductive polymers (thermoplastics, thermosets, and elastomers) has been considered as a possible strategy to obtain conductive polymer-based (nano)composites with expected enhanced EMI shielding performances. Hence, a great number of researchers have considered developing such polymer-based (nano)composites by creating lightweight structural–functional components with improved electrical conductivities, with the intention of improving EMI shielding efficiencies for the most varied applications; this is especially pertinent for highly demanding sectors, such as aerospace or telecommunications [1,2]. Although promising, as the properties (including the electrical conductivity, and as a result, the EMI performance) of polymer (nano)composites may be refined with the addition of functional conductive nanofillers (type(s), concentration, method of preparation, etc.), the possible advantages of foaming such materials are undeniable [3,4]. Not only does foaming lead to even lighter structures, with direct advantages in fields such as aerospace, and enhanced functional characteristics such as reduced thermal conductivity, but it has also been proven to effectively reduce the percolation threshold of the conductive nanofillers, enabling lightweight components to be obtained, with high electrical conductivities at lower concentrations of nanofiller. Additionally, in the specific case of EMI shielding, the generation of a controlled cellular structure, especially of the microcellular type, promotes the absorption/multiple reflection-based EMI shielding mechanism. In doing so, the common problem concerning EMI shielding components based on reflective materials (such as metals), and the manner in which they interfere with the reflected waves and surrounding electronic components, is mitigated [3,5,6]. In sum, combining the addition of conductive nanofillers with polymers, and generating a controlled cellular/porous structure, enables the creation of lightweight components with enhanced electrical conductivities and EMI shielding properties. Therefore, this review article addresses the most recent trends in polymeric foams and porous structures intended for EMI shielding applications in the last three years. Initially, this review will consider the importance and the possibility of generating foams with a controlled microcellular structure as a way to enhance EMI shielding and promote an absorption-based EMI shielding mechanism, focusing on both common, as well as advanced, technologies/methods that use sCO_2_. Lesser known, but equally important studies are also considered in this review, including works that use polymer foams as templates to create carbon foams for high temperature EMI shielding applications. Additionally, as the generation of a controlled cellular structure may affect the EMI shielding performance of the prepared foams (these also contain conductive nanofillers), this review analyzes the importance of the type(s) of nanofiller(s) used, the effective possible combination of different types of nanofillers (hybrid nanofillers/nanohybrids), and the selective distribution of said nanofillers in the formation of a highly conductive network at lower nanofiller concentrations; this results in the enhancement of the EMI SE. Works dedicated to the use of computational tools that assess and tailor the EMI shielding performance of polymer foams (represented in this study by a recently published work) are not as well-known as studies which focus on the influence of the cellular structure and the generated polymer–nanofiller(s) microstructure on the EMI shielding performance of polymer nanocomposite foams, though this topic nevertheless attracts a high level of interest. The last section further expands upon the concept of polymer-based foams to include other porous polymer structures, with a focus on porous components that were prepared using 3D printing; this is expected to be highly applicable in a great number of fields, as 3D printing technologies have undergone a considerable technological boost in recent years, and this trend is expected to continue in the coming years. A summary of the recently published works (concerning polymeric foams and polymer-based porous structures for EMI shielding) that are considered in this review is presented in Table 1, and it includes key innovation points. Finally, some future perspectives are presented and categorized based on material/component development, the use of advanced computational predictive tools based on Artificial Intelligence (AI) and Material Learning (ML) approaches, and more application-driven approaches are also considered. A summary of the recent strategies and future perspectives to enhance the EMI shielding performance of polymeric foams and porous structures is presented in Figure 1.

## 2. Microcellular Foaming for EMI Shielding Applications

### 2.1. Supercritical CO_2_ Foaming

As several researchers have demonstrated the importance of developing foams with a controlled microcellular structure for promoting absorption-dominant EMI shielding, sCO_2_ foaming has been well regarded as a foaming strategy for the fabrication of lightweight materials intended for highly demanding EMI shielding protection applications [9,30,31]. Indeed, Dun and co-workers [7] prepared solid state sCO_2_ foaming polyvinylidene difluoride-carbon nanotubes (PVDF-CNT) using nanocomposite foams for EMI shielding purposes. They demonstrated that the generation of a cellular structure during foaming gradually favored the interconnection between carbon nanotubes, thus promoting the formation of an effective, conductive CNT network through the cell walls of the foams. As a consequence, the final materials comprised high electrical conductivities and EMI shielding efficiencies (reaching a specific value of almost 30 dB·cm^3^/g). Additionally, as the generation of the cellular structure during sCO_2_ foaming led to the reorientation of CNTs, the resultant foams displayed considerably lower percolation thresholds for electrical conduction when compared with their un-foamed nanocomposite counterparts (0.40 vol% CNT compared with 1.44 vol% CNT, i.e., almost four times lower). The authors also demonstrated how the EMI shielding mechanism gradually changed from being reflection-dominant in the case of the un-foamed nanocomposites, to absorption-dominant with foaming; the controlled cellular structure also contributed to an effective mechanism whereby multiple reflections occurred (see scheme presented in Figure 2).

Aghvami-Panah et al. [8] used microwave-assisted foaming from previously sCO_2_ saturated precursors to prepare polystyrene (PS)-based nanocomposite foams containing different concentrations of three different types of carbon-based (nano)fillers, as follows: carbon black (CB), carbon nanotubes (CNTs), and graphene nanoplatelets (GnP). Then, they analyzed the EMI shielding performance of the resultant foams. They demonstrated that the type of (nano)filler and its concentration has important effects on the final cellular structure, and hence, on the properties of the resulting foams, with the maximum value of the specific EMI SE being obtained for PS foams containing a 1 wt% of CNTs (>50 dB·cm^3^/g).

Wang et al. [9] prepared poly(methyl methacrylate) (PMMA)-based microcellular foams containing different graphene oxide (GO)-nickel nanochains (NiNCs); the second type of nanochain comprises known electric and magnetic anisotropic characteristics, using a sCO_2_ foaming process. The combination of the two types of particles, with different dimensionalities (1D NiNCs and 2D GO nanosheets), favored the formation of an effective 3D conductive filler network, resulting in foams with higher electrical conductivities at lower filler concentrations, and absorption-dominated high EMI SE, in some cases surpassing 53 dB.

In addition to some of the previously mentioned environmental advantages of creating lightweight components with enhanced EMI shielding performances for highly demanding applications via foaming (for instance, the use of a considerably lower amount of polymer to create the component), there is considerable interest in further improving the sustainability of these materials by using recycled and recyclable polymers. Recently, Lee et al. [10] developed nanocomposite foams with outstanding EMI shielding protection by integrating chemically recycled poly(ethylene terephthalate) (PET) (notably, a thermoplastic polyamide elastomer synthesized with a monomer derived from the aminolysis of recycled PET) with single wall carbon nanotubes (SWCNTs), and later, microcellular foaming, using sCO_2_ dissolution (see scheme presented in Figure 3). The authors showed that by adding only 2 wt% of SWCNTs, the resultant foams displayed extremely high EMI shielding efficiencies; the specific values exceeded 210 dB·cm^3^/g. As well as showing great durability, the nanocomposite foams could be easily recycled, reprocessed, and even re-foamed (see Figure 4), thus providing promising avenues for reducing plastic waste accumulation and reducing the necessity of virgin plastics in the production of foams for sustainable electronic applications.

### 2.2. Other Microcellular Foaming Technologies

In addition to sCO_2_ foaming, there are other manufacturing technologies that can be used to create microcellular polymer-based foams [31], from more traditional extrusion processes and injection-molding microcellular foaming [32,33,34,35], to more advanced and recent processes such as ultrasound-aided [36,37,38], bi-modal [39,40,41,42], or cyclic [43,44] microcellular foaming, which included solvent-based processes such as phase separation/inversion processes [45]. Hence, some research groups have considered these to be possible methods for developing microcellular lightweight components with enhanced absorption-dominated EMI shielding characteristics. Such is the case of Zhu et al. [11], who recently considered the preparation of microcellular polyamide 6 (PA6)-carbon fiber (CF) composite foams by means of chemical injection foaming. As previously mentioned, the combination of CF addition and chemical foaming led to the production of foams with more uniform, smaller, cellular structures. Consequently, the EMI SE produced comparatively higher results than the un-foamed composite counterpart (around 37 dB, almost 30% higher than the non-foamed composite). A similar injection molding-based foaming process (specifically, the core-back injection process) was used by Wang and co-workers [12] to prepare lightweight polypropylene (PP)-carbon nanosheets (CNS) and nanocomposite foams with microcellular structures. In this work, the authors showed that it was possible to obtain foams with higher specific flexural modulus values than that of their respective un-foamed injection-molded counterparts; hence, foams show great potential as EMI shields for sensor-based applications.

## 3. Carbon Foams Obtained from Polymer Foam Templates

Interestingly, researchers have recently considered strategies concerning the synthesis of carbon foams, which are known for their high stiffness and intrinsic ability to be used at high temperatures (they can be used in oxidative surroundings with temperatures higher than 500 °C, and in inert environments with temperatures up to 3000 °C [46]), as well as their enhanced EMI shielding performances which result from polymeric foam/porous templates. Due to these traits, they could find potential applications in the aerospace and telecommunication fields as structural–functional integrated elements. As such, Li et al. [47] prepared carbon foams via the carbonization of previously prepared thermosetting polyimide (PI) foam templates (see Figure 5). The resultant foams displayed a combination of good mechanical performance (high compressive strength) and high EMI shielding efficiency, reaching values as high as 54 dB (593.4 dB·cm^3^/g in terms of specific EMI SE), which were due to the crosslinked network of polyimide templates. The authors also showed that it was possible to adjust the density, compressive mechanical performance, and electrical conductivity (and hence, the EMI SE) by carefully controlling the characteristics of the polyimide templates. Similarly, Sharma and co-workers [48] prepared lightweight carbon foams by carbonizing phenolic-impregnated polyurethane foam templates, and they later decorated the foams with zinc oxide nanofibers by means of electrospinning. They observed that the characteristic open-cell porous structure of the PU foam templates, together with the presence of the nanofibers, results in lightweight components with absorption-dominant enhanced EMI shielding performances (>58 dB and >1000 dB·cm^3^/g for a final density as low as 0.28 g/cm^3^). 

Tang et al. [49] used some of the previously mentioned strategies, including a combination of prepared emulsion-templated, open-cell polymeric composites, which were crosslinked (hypercrosslinked) and carbonized, to fabricate lightweight carbon foams from syndiotactic polystyrene, as well as carbon nanotubes (CNTs) and magnetic Ni-CNTs. The additional production of both CNTs and magnetic Ni-CNTs helped to better control the carbonization stage when preparing the carbon foams, leading to less shrinkage, a higher graphitization degree, larger cells, and nano-reinforced pore walls. As a result, the prepared carbon foams displayed high mechanical robustness, a more hydrophobic nature, and a higher specific surface area (as high as 144 m^2^/g). The simultaneous use of CNTs and Ni-CNTs nanofillers greatly enhanced the EMI SE, reaching values of almost 80 dB for some of the prepared carbon foams. The authors explained this great EMI SE improvement by noting the electrical conduction losses within the conductive carbonized PS-carbon-rich nanofiller network, the polarization losses from the interfacial polarization between the carbonized PS and the nanofillers, dielectric losses within the cells, and magnetic loss from the magnetic Ni-CNTs.

Some researchers have recently addressed the issues concerning both EMI protection and thermal management during the operation of electronic components by combining carbon foams containing conductive carbon-based nanofillers (such as GO, rGO or carbon nanotubes) with phase change materials. For instance, Gao and co-workers [50] prepared carbon foams with rGO-containing, paraffin-based, phase change materials (see Figure 6), with improved EMI SE and thermal conductivity. The authors attributed this improvement to the effective construction of an electrically and thermally conductive, carbonized melamine foam-rGO 3D network (Figure 7). In particular, the authors were able to prepare lightweight components with an EMI SE of up to 49 dB, absorption-dominated EMI behavior, and a thermal conductivity that was more than 300% higher than that of pure paraffin, enabling a reduction in the working temperature of an analog electronic chip device of more than 17 °C. Moreover, the developed components exhibited high cyclic and thermal stability, ensuring their long-term use in these sorts of electronic devices.

## 4. Hybrid Nanofillers/Nanohybrids

As with polymer-based (nano)composites, researchers have recently considered the possibility of tailoring the EMI shielding efficiency of cellular/porous polymer-based (nano)composites, not only by controlling their cellular structure (void fraction, open/closed/interconnected cellular structure, cell size, etc.), but by favoring the formation of an effective, electrically-conductive network using the solid fraction. They achieved this by combining different types and proportions of nanofillers (hybrid nanofillers or nanohybrids), as there is a direct relationship between electrical conductivity and the EMI SE. As such, Dehghan et al. [13] prepared cellular nanocomposites using sCO_2_ and hybrid MXene/rGO nanosheets, showing that the use of the hybrid fillers led to foams with smaller cell sizes and higher values of electrical conductivity when compared with those containing only MXene; overall, this resulted in components with higher EMI shielding efficiencies (>25 dB for foams containing MXene/rGO nanohybrids, and less than 18 dB for foams containing only MXene (see comparison between foams containing only MXene and foams containing MXene/rGO nanohybrids in Figure 8)). Zhang et al. [14] combined carbon-based nanoparticles with different geometries, particularly those comprising graphene layer-like nanosheets and carbon nanotubes, and they prepared microcellular, PMMA-based, (nano)composite foams containing these nanohybrids via sCO_2_ dissolution. Interestingly, they demonstrated that foaming promoted the exfoliation of the graphene nanosheets and the better distribution of the nanohybrids throughout the solid fraction of the foams; the CNTs connected the graphene sheets, which helped to create a highly effective conductive 3D network, and led to the production of materials with high electrical conductivities and EMI shielding efficiencies, especially when compared to their un-foamed counterparts.

Cheng and co-workers [15] took advantage of the unique structure of some types of polymeric foams, including the particular sponge-like structure of some waterborne polyurethane (PU) foams, by preparing dip-coated, PU-based, composite foams containing CNT-magnetic anisotropic Ni particle hybrids that were intended to be used for EMI shielding and protective purposes. Once again, the authors demonstrated the effective synergistic effect of combining both types of nanofillers, as the foams containing the hybrids reached EMI SE values that were greater than 42 dB, clearly exceeding those of the PU foams containing only CNTs. Based on the results, the authors proposed a mechanism to describe the absorption-dominated EMI shielding characteristics of the resultant PU-CNT-Ni foams (see Figure 9). First, the EM waves were attenuated due to conduction losses in alternating EM fields, which were promoted by the electrically conductive solid fraction; second, the addition of the Ni particles made the EM wave impedance match with that of the composites, with interfacial polarization losses between the CNTs, Ni, and PU, which thus caused wave absorption; and third, the attenuation of EM waves was also promoted by the magnetic losses caused by the magnetic Ni particles. Additionally, incidental EM waves were reflected and scattered multiple times at the wall interface, which was formed by the hybrid filler structure, thus further dissipating EM energy. Zheng et al. [16] used a similar PU-based foamed system which combined Fe_3_O_4_-polyvinyl acetate (PVA) and GO-silver particles for use as a (nano)filler, thus demonstrating the possible synergistic effects of both types of (nano)fillers in terms of establishing an effective network using the PU foam structure; this led to high EMI shielding efficiencies (>30 dB and almost 280 dB·cm^3^/g) and an absorption-dominated primary shielding mechanism. As with the use of Ni, the addition of the ferromagnetic Fe_3_O_4_ particles effectively improved the absorption performance due to the combined characteristic dielectric loss of ferroelectric materials and the hysteresis loss of ferromagnets. Zhu et al. [17] were able to modulate the EMI shielding properties of polymer composite foams that were prepared by double dip-coating a previously prepared sponge-like PU foam by first making it conductive in a SWCNT dispersion, and then coating the already conductive foam with a paraffin solution/emulsion (see Figure 10); this led to the creation of a shaped, memory-like foam, the EMI shielding performance of which may be modulated by applying a controlled compressive stress for later possible recovery. The resultant foams showed extremely high values for the EMI SE (56 dB) at ultra-low densities (0.03 g/cm^3^) after adding SWCNTs content below 0.2 vol%; they displayed ultra-high durability (>2000 compression-recovery cycles) and adjustable, continuous, EMI shielding efficiencies between 18 and 30 dB.

As a consequence of their characteristic mechanical performance, lightness, and enhanced EMI shielding protection, these foams are expected to find possible applications in the field of electronics, for instance, in portable and wearable electronic devices.

## 5. Selective Distribution of Conductive Nanofillers

Wang et al. [18] recently used an interesting strategy to enhance the EMI shielding performance of thermoplastic polyurethane (TPU)-CNT composite foams. This strategy required creating a multilayer gradient structure (see scheme of the preparation process in Figure 11) by adjusting the cellular structure and ratio of the soft/hard domains of each layer, combined with the selective distribution of conductive carbon nanotubes throughout the hard domain (see scheme presented in Figure 12a). This selective distribution of the conductive CNTs improved the interlayer polarization of the EM, and the impedance matching between the material and air, leading to the production of foams with EMI SE (>35 dB) results that were 20% higher than that of the homogenous composites containing the same amount of CNTs. The proposed EMI shielding mechanism of the TPU–CNT composites with multilayer gradient structures after foaming (presented in Figure 12b), were based on a combination of polarization losses and multiple reflections. This enabled the researchers to correlate the importance of selectively distributing conductive nanoparticles throughout a given domain of the material with the EMI shielding performance, thus opening new possibilities when designing lightweight components that are based on low filler composites with enhanced EMI shielding protection. Some researchers have even further extended this layered, gradient-like concept, combining conductive foamed layers that are based on a polymer nanocomposite foam, with layers formed by different materials, such as aluminum, knitted fabric layers, etc., thus endowing the sandwich-like materials with excellent EMI shielding [51].

Similarly to Wang et al., Liu and co-workers [19] prepared biodegradable polycaprolactone (PCL)/poly(lactic acid) (PLA)-CNT composite foams by selectively distributing/dispersing the CNTs throughout the PCL matrix; as such, they facilitated the formation of an effective conductive network in the final prepared foams. As a result, the resultant foams displayed absorption-dominant EMI shielding behavior with an EMI SE of almost 23 dB and a specific EMI SE of around 90 dB·cm^3^/g; this provides new possibilities for the development of fully biodegradable components for EMI shielding protection.

Interestingly, Feng et al. [20] were able to selectively distribute conductive CNTs at the interfaces of polyetherimide (PEI) granules when preparing PEI-CNT foams through ball milling, sinter molding, and later, the sCO_2_ dissolution foaming process; this led to the formation of conductive paths at extremely low CNT concentrations (percolation threshold of 0.06 vol%), and hence, to foams with high values of electrical conductivity (almost 8 S/m) and high EMI shielding efficiencies (>30 dB).

Taking advantage of the fact that the generation of a controlled and particular cellular structure during foaming may reduce the actual conductive paths between conductive particles pre-dispersed throughout the polymer-based matrix before foaming due to an excluded volume effect and even promote the redistribution and alignment of the particles, Wu and co-workers [21] were able to prepare PP-CNT composite foams with enhanced EMI shielding efficiencies (reaching values of almost 60 dB) using a core-back foaming injection molding process, as foaming enabled the creation of cellular structures with big cells, which promoted the formation of a more effective path comprising aligned conductive particles. Yang et al. [22] had already used a similar strategy to prepare highly conductive polymer-based composites at extremely low rGO percolation thresholds (0.055 vol%). Peng and co-workers [23] promoted the formation of a bi-conductive network in polymer-based foams by combining the high efficiency of Ni-coated melamine foam sponges with absorption-driven EMI shielding elements, in addition to the redistribution and arrangement of conductive CNTs induced by foaming in CNT-TSM-PDMS composite foams; this led to a high EMI SE value of 45 dB for CNT contents of only 3 wt%. As expected, EMI shielding was absorption-dominated, and the foams also showed outstanding EMI shielding durability, even in harsh environments.

## 6. Computational Approaches

Owing mostly to their (micro)structural complexity, the scientific works that deal with polymer (nano)composites, polymer foams, and polymer-based (nano)composite foams intended for EMI shielding protection, where the EMI shielding mechanism is absorption dominated, still predominantly rely on trial and error. They do not consider, or they only superficially mention, the real mechanisms governing the (micro)structure–EMI shielding performance relationship of these materials. Only recently have researchers started to consider the use of computational-based approaches to develop composite foams for EMI shielding. In particular, Park et al. [52] have examined the fabrication of layered PVDF-based composite foams, focusing on the effects of the layered structure and microcellular structure of the developed layered foams on their final EMI shielding performance. Through the use of a theoretical computation approach, they demonstrated that the best result (for layered structures) consisted of an absorption layer based on a medium-density PVDF foam containing a high amount of SiC nanowires and MXene nanosheets, as well as a shielding layer formed by a lower density PVDF-8 wt% CNT nanocomposite foam (see Figure 13). Experimental findings also corroborate the importance of optimizing the void fractions of both foamed layers so as to guarantee the best possible input impedance matching. Moreover, the addition of nanohybrids, based on 2D conductive MXene nanosheets and 1D SiC nanowires in the PVDF, in the absorption layer, provided significant EM wave dissipation enhancements. Hence, the tailored, layered structures exhibit a combination of low EM reflectivity and high EMI SE, providing significant possibilities regarding the development of tailor-made elements, with absorption-dominant EMI shielding characteristics, for a vast array of high performance applications.

## 7. 3D Printing

In 2018, Bregman and co-workers [24] published an article where they presented a model-based methodology (finite element modeling in COMSOL) for estimating the EMI shielding properties of polymer-based composites, using measurements of complex EM parameters, predictive absorption modeling, and later, fabrication by 3D printing. More specifically, they used fused filament fabrication and compression-molding. Some of the presented preliminary results were derived from a commercial PLA-based filament containing 16 wt% of graphene/carbon nanotubes/carbon black conductive filler, which suggests that the presence of a controlled cellular structure (periodic pore structure) in a conductive polymer (nano)composite can lead to lower EM reflection losses and higher absorption capabilities. The authors claimed that this was possible due to a combination of dielectric mismatches between the conductive PLA-based filament and the air present inside of the pores, which caused multiple reflections at the interfaces. Additionally, the presence of dissipation pathways (conductive fillers), as well as a material fraction whose relative impedance is 1 or close to 1 (air inside of pores), was associated with a lower front face reflection, and hence, higher absorption.

Using a similar fused deposition modeling (FDM) 3D printing process (see scheme presented in Figure 14), Lv et al. [25] prepared polyolefin elastomer (POE)-graphene nanoplatelet (GnPs) nanocomposite foams and demonstrated their viability for thermal management applications as a result of their enhanced absorption-based electromagnetic interference shielding efficiency (EM mixed multiple reflection-absorption mechanism can be seen in Figure 15). These foams produced values as high as 35 dB, or almost 245 dB·cm^2^/g (thickness-normalized specific value), and they exhibited high thermal conductivity (4.3 W/(m·K)).

Shi and co-workers [26] prepared FDM 3D-printing honeycomb-like cellular nanocomposites based on a combination of PLA and conductive GnP/CNT nanohybrids (see images presented in Figure 16). The resultant cellular/porous parts showed improved mechanical performances and EMI shielding efficiencies as high as 37 dB (X-band range); this is significantly higher than the common commercial standard of 20 dB for EMI shielding applications (see Figure 17). Interestingly, the authors employed a finite element simulation, based on the fluid dynamics of polymer melts, to optimize the amount of GnP and CNT in the nanohybrids (optimum values of 2 and 4 wt%, respectively), and they established a quantitative relationship between the final cellular/porous structure and the EMI shielding properties of the 3D printed component, thus facilitating the design of novel, 3D printed, lightweight structures for electromagnetic radiation protection.

Similarly, but counteracting one of the main problems associated with adding conductive carbon-based nanofillers (in this case, carbon nanotubes) to a polymer matrix (which is nanofiller agglomeration), Pei et al. [27] used high energy mechanical ball milling to evenly distribute/disperse CNTs throughout a chitosan (CS) matrix and create CS-CNTs 3D printing ink. After creating porous parts via 3D printing (see Figure 18), the authors demonstrated that they effectively absorbed EM waves, reaching EMI shielding efficiencies higher than 25 dB, and high absorption losses (>19 dB) for densities as low as 0.072 g/cm^3^, hence showing that the prepared ink may find promising applications in the 3D printing of lightweight components with improved EMI shielding performances.

More recently, some authors have considered 3D printing polymer-based aerogels for EMI shielding and piezoresistive sensor applications. Indeed, Guo and co-workers [28] have developed a strategy based on the 3D printing of lamellar graphene aerogels (LGA). More specifically, they have conducted printing through a slit extrusion printhead of a shear, thinning, GO water-based dispersion (see Figure 19). As they were able to better control the size and shape of the graphene platelets compared with more traditional methods, the LGAs showed higher EMI shielding efficiencies (up to almost 69 dB for a 3 mm thickness).

Xue et al. [29] took this concept further, and via 3D printing, they prepared aerogel frames with hierarchical porous structures; hence, gradient-conductivity due to the MXene-CNT-Polyimide occurred. This shows that the prepared aerogels almost displayed full absorption-based EM behavior, due to the slightly conductive EM absorptive top layer, and the highly conductive EM reflective bottom layer, combined with the dissipation of EM waves caused by multiple reflections (hierarchical porous structure). As a result, aerogels with extremely high EMI shielding efficiencies (>68 dB) were created, and they exhibited extremely low reflection losses; therefore, they might have promising applications in the defense and aerospace industries.

## 8. Future Perspectives and Concluding Remarks

The continued development and application of polymer-based (nano)composite foams and porous structures for EMI protection (which are currently highly topical and needed due to the technological challenges that are associated with the increasing use of wireless communication electronic devices), still require a great deal of work. Indeed, further research is required on the (micro)structural complexity of such materials, in addition to the technological challenges associated with their preparation as there are increasing demands being made by the electronics industry, which now requires components with EMI shielding protection and a vast range of additional properties, such as heat dissipation or long-term stable mechanical performance. Future directions for the development and application of such materials may be described in terms of three parallel paths, as follows: the first is related to the development of novel materials/components (i.e., their composition, the development of new processes/methods/technologies to carefully control their cellular structure (foams as well as porous structures)), and the design and use of gradient/layered structures; the second is related to the design of such materials using AI and ML approaches; and the third is more application-driven, with the possible development of strategies to tailor and control the (micro)structural characteristics of the developed foams/porous structures, and hence, their properties, for specific applications.

The first of the abovementioned paths is related to the development of new polymer-based formulations and components, and it has already been mentioned to a great extent in this review, as it initially focuses on the adaptation of the composition of the polymer nanocomposite foams to the foaming process and to the required final characteristics of the components to be used for EMI shielding. The most common strategy, which is expected to be further explored in the coming years, is the addition to polymer matrices and proper distribution/dispersion of hybrid fillers that combine functional nanoparticles, especially conductive and/or magnetic nanoparticles with different geometries/dimensionalities/characteristics, such as tubular-like carbon nanotubes and flat-like graphenes. Chemically, polymer matrices with reactive functional groups may also be modified using functional molecules and/or (nano)fillers. Even more specific compositional modifications can be considered, in conjunction with the previously mentioned selective distribution/dispersion/preferred orientation of the nanofillers, which involve the addition of nucleating agents to promote cell size reduction in the foams and the crystallization in semi-crystalline polymer matrices [53], or the control of the phase morphology of the polymer-based solid fraction of the foam (for instance, sCO_2_ annealing [54] could be used as a strategy to enhance the EMI SE). Moreover, regarding more application-driven future research directions, adding functional (nano)fillers to bio-based polymers and foams might be possible, so as to create biodegradable lightweight components for more short/medium-term EMI shielding applications.

In addition to composition, the development of new fabrication processes and technologies that generate controlled cellular/porous structures which are intended to maximize EMI shielding are especially focused on favoring EM wave absorption; this is critical in terms of enabling the transfer of scientific knowledge to the industry. In this sense, new industrial foaming technologies and fabrication processes are expected to arise in the future, especially those intended to generate microcellular foams and advanced microcellular foaming processes such as ultrasound-assisted sCO_2_ foaming or multi-modal and cyclic foaming processes, or controlled porous structures such as 3D printing.

Finally, the design and use of gradient/layered structures, where at least one of the layers is formed by an EM wave absorber material based on a polymer-based foam or porous structure (previously mentioned in this review), is expected to be one of the most interesting strategies in terms of developing lightweight components with a tailor-made EMI shielding performance.

As mentioned previously in Section 6, only recently have researchers started to use computational approaches to design composite foams for EMI protection. This is a topical field of interest that has received a boost in the last 2–3 years due to the possible use of powerful predictive tools for data-driven multiphase systems based on polymer-based nanocomposites such as AI and ML; these methods greatly surpass the capabilities of traditional computational approaches [55]. Different ML algorithms, including Artificial Neural Network (ANN), Adaptive Neuro Fuzzy Interference Systems (ANFIS), and deep learning Convolutional Neural Network (CNN), among others, as they train using enormous amounts of experimental data, have been demonstrated as very powerful and useful predictive tools for polymer nanocomposites; indeed, they provide unique insights and much deeper explorations into the properties of such complex multiphase materials, including EMI shielding [56,57]. Nevertheless, as they strongly correlate with the quantity and quality of the initial experimental data, they are still somewhat limited for use with polymer nanocomposites, as a limited number of specific databases have been built, NanoMine being one of them [58]. This limitation is expected to be solved in the coming years using tools which extract the appropriate scientific data from reliable scientific sources using ML-based natural language processing techniques; conversely, advanced simulation approaches might be used, such as multiscale modelling [55]. Other interesting options are also being explored, such as hybrid machine learning, which combines multiple ML algorithms, and hence, which increase the prediction capability [59]; alternatively, adaptive learning might be used, which involves reinforcement learning based on the information gathered from the surrounding working environment [60]. Even more scarce is the application of AI and ML-based tools to the analysis of polymeric foams and porous structures; indeed, they are rarely used to enhance the understanding of the effect of the cellular/porous structure development on EMI shielding mechanisms and efficiencies. As such, Shi and co-workers [61] recently combined deep convolutional neural network (DCNN) visualization with experimental data to analyze the EMI shielding of porous PVDF-CNT nanocomposites; this involved a modified deep residual network (ResNet) which was trained using SEM micrographs of the prepared foams, demonstrating its effectiveness in assessing the impact of the cellular structure (cell size and cell density) on the shielding mechanisms, with bigger cell sizes contributing less to EM wave absorption. This study, alongside others that have focused on the preparation of porous structures created by 3D printing [62], proves how AI and ML approaches can significantly help study complex multiphase materials, particularly the analysis of properties that are greatly dependent on material (micro)structures, and which are affected by different mechanisms, as in the case of EMI shielding. Therefore, AI and ML are expected to be increasingly used in the coming years when designing components based on polymeric foams or porous structures for EMI protection.

As expected, some of the future trends in polymer-based (nano)composite foams and porous structures are more application-driven, and they try to connect the best of both scientific and industrial worlds. There is a significant interest in developing viable strategies to control, and if possible, tailor the characteristics of the resultant foams, so as to enable the preparation of lightweight components with adjustable EMI shielding properties for specific industries and applications, from flexible foams for sensors to rigid porous structures for highly demanding structural applications.

## Figures and Tables

**Figure 1 polymers-16-00195-f001:**
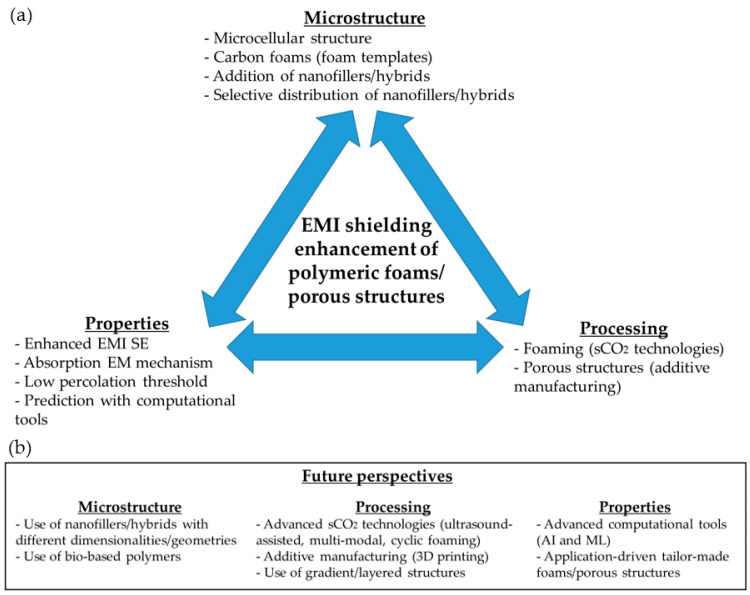
(**a**) Recent strategies used to enhance the EMI shielding properties of polymeric foams/porous strutures and (**b**) future perspectives.

**Figure 2 polymers-16-00195-f002:**
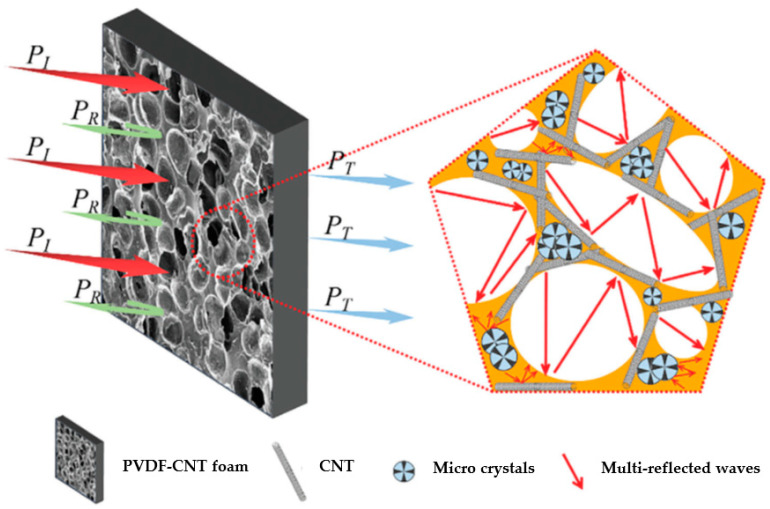
Scheme of EM wave dissipation in PVDF-CNT foams [7]. Copyright 2023, John Wiley & Sons. Note: *P_I_*—incident wave power; *P_R_*—reflected wave power; *P_T_*—transmitted wave power.

**Figure 3 polymers-16-00195-f003:**
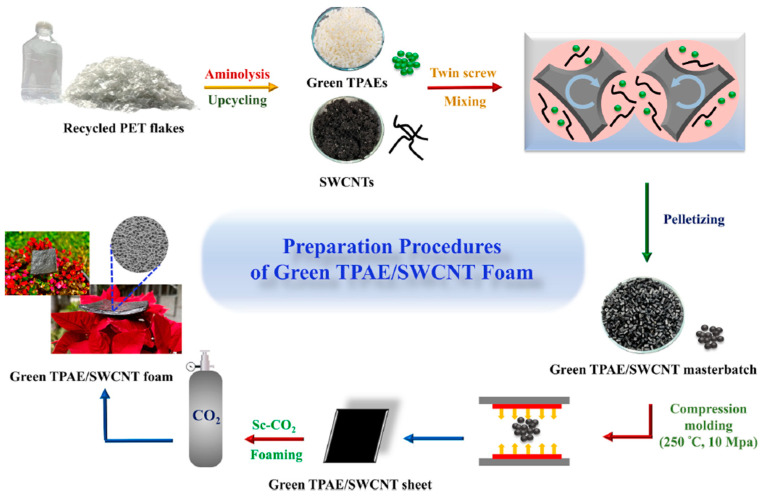
Procedure for preparing green TPAE-SWCNT nanocomposite foams [10]. Copyright 2023, Elsevier.

**Figure 4 polymers-16-00195-f004:**
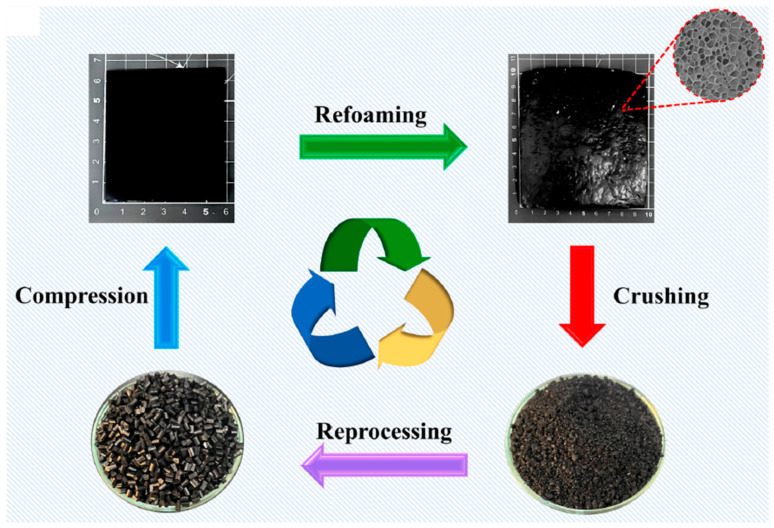
Recycling, reprocessing and refoaming of TPAE-SWCNT nanocomposite foams [10]. Copyright 2023, Elsevier.

**Figure 5 polymers-16-00195-f005:**
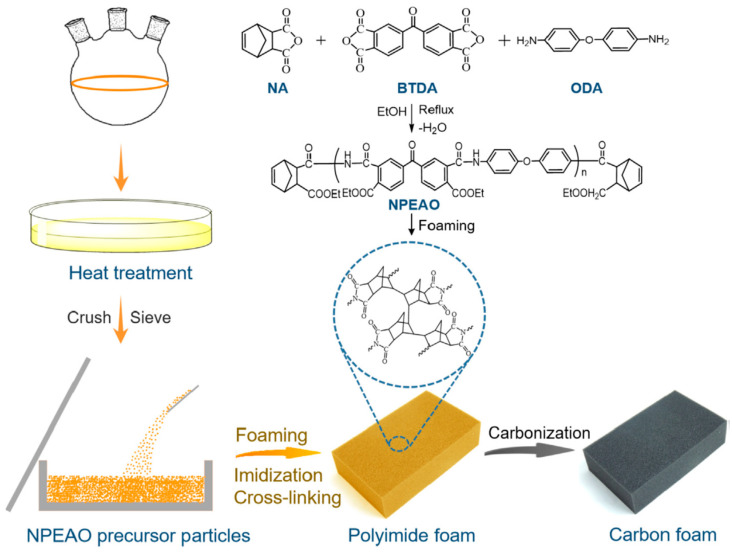
Scheme showing the fabrication of carbon foams using thermosetting PI foam [47]. Copyright 2023, Elsevier.

**Figure 6 polymers-16-00195-f006:**
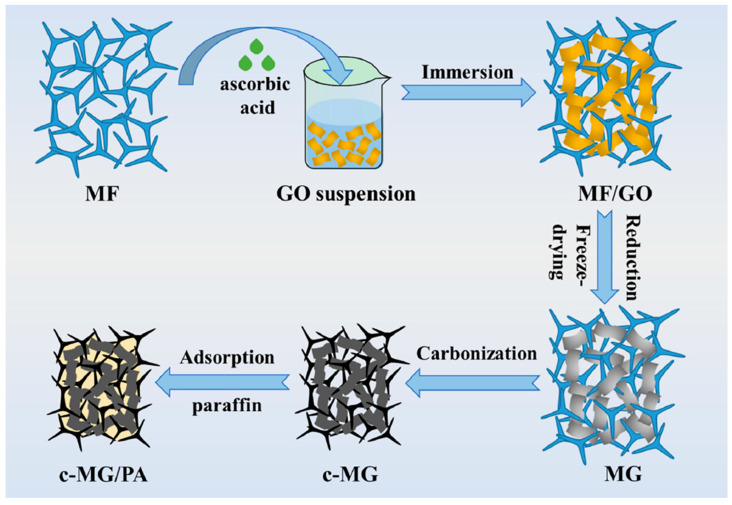
Scheme showing the fabrication of c-MG/PA [50]. Copyright 2023, Elsevier.

**Figure 7 polymers-16-00195-f007:**
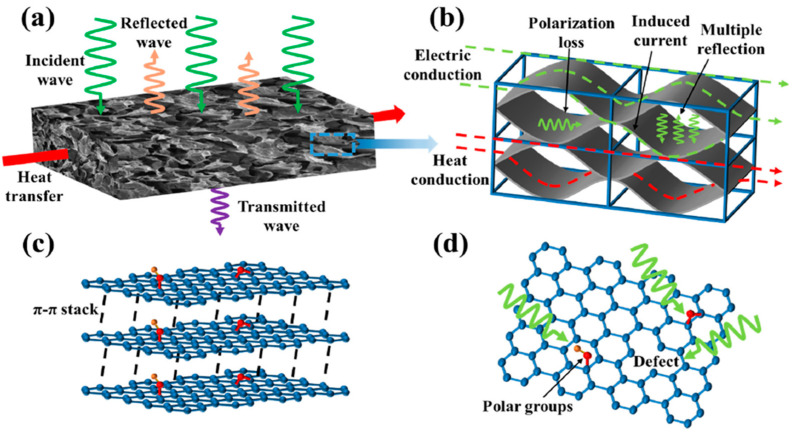
(**a**) Electromagnetic shielding and heat transfer diagram of c-MG/PA; (**b**) electromagnetic shielding and heat transfer mechanism of the dual cross-linking network of c-MG; (**c**) interface polarization loss mechanism of the GnP network; (**d**) dipole polarization loss mechanism of rGO network [50]. Copyright 2023, Elsevier.

**Figure 8 polymers-16-00195-f008:**
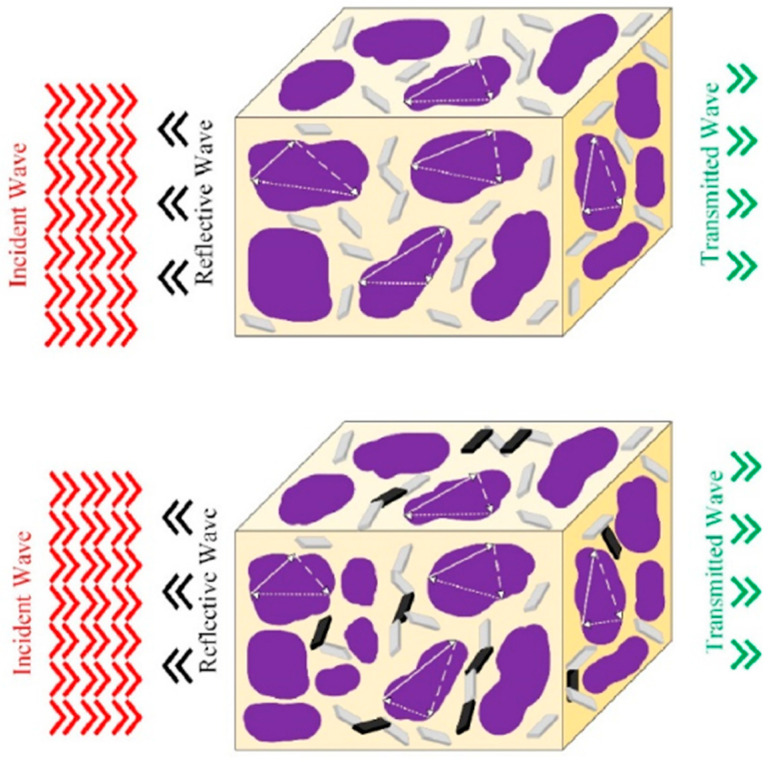
Scheme of EM waves passing through the cellular structure of a foam containing MXene and rGO [13]. With permission from Elsevier. Legend: red, black, and green arrows represent incidental, reflective, and transmitted waves, respectively; gray and black layers indicate MXene and rGO, respectively.

**Figure 9 polymers-16-00195-f009:**
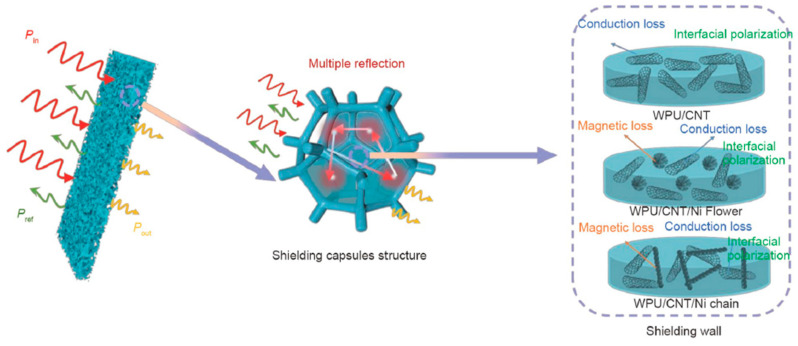
Scheme of the EMI shielding mechanism for PU-CNT and PU-CNT-Ni composites [15]. Copyright 2023, Springer Nature.

**Figure 10 polymers-16-00195-f010:**
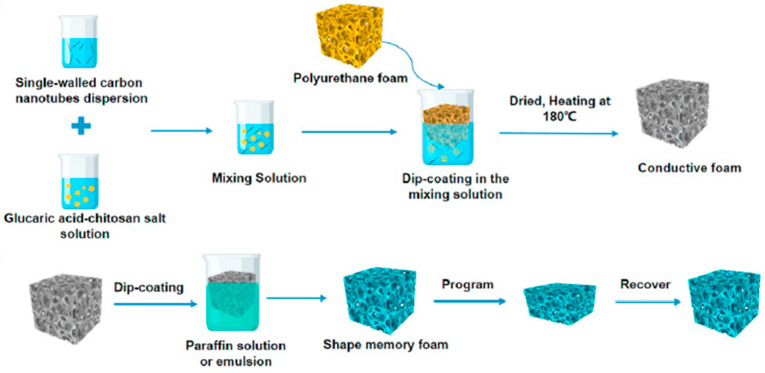
Scheme of the fabrication process for conductive and shaped memory PU foams for EMI shielding [17]. Copyright 2023, Elsevier.

**Figure 11 polymers-16-00195-f011:**
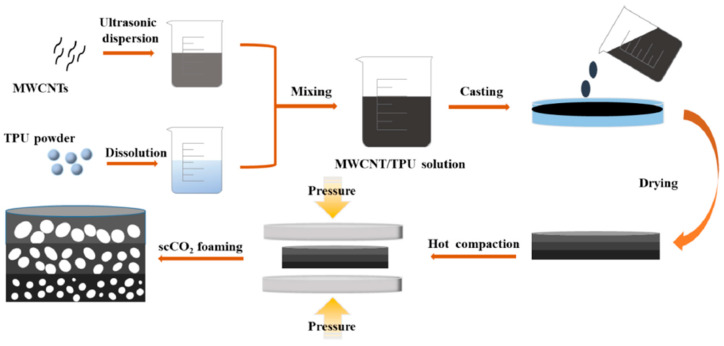
Scheme showing the preparation of TPU-MWCNTs foam with a multilayer gradient structure [18]. Copyright 2023, Elsevier.

**Figure 12 polymers-16-00195-f012:**
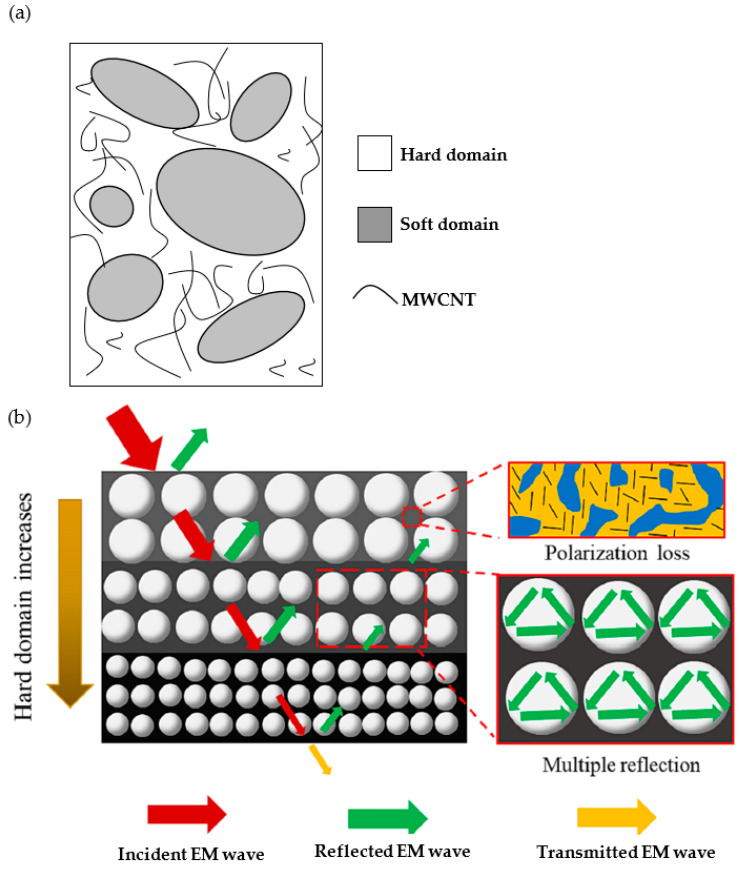
(**a**) Scheme showing the phase separation of TPU-MWCNTs systems (adapted from [18]) and (**b**) representation of the EMI shielding mechanism of TPU-MWCNTs foam with a multilayer gradient structure [18]. Copyright 2023, Elsevier.

**Figure 13 polymers-16-00195-f013:**
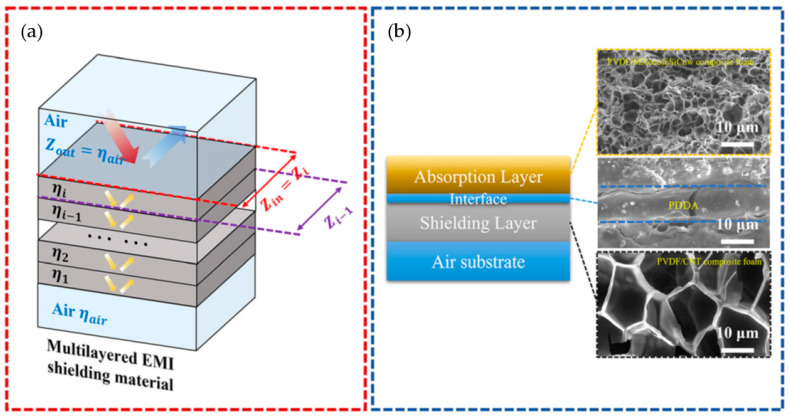
(**a**) Representation of the input impedance of the multilayered EMI shielding material with i layers, and (**b**) scheme showing the layered composite foam structure formed by PVDF-30 wt% SiCnw@MXene10:1 composite foam as the absorption layer, and PVDF-8 wt% CNT composite foam as the shielding layer [52]. Copyright 2023, Royal Society of Chemistry.

**Figure 14 polymers-16-00195-f014:**
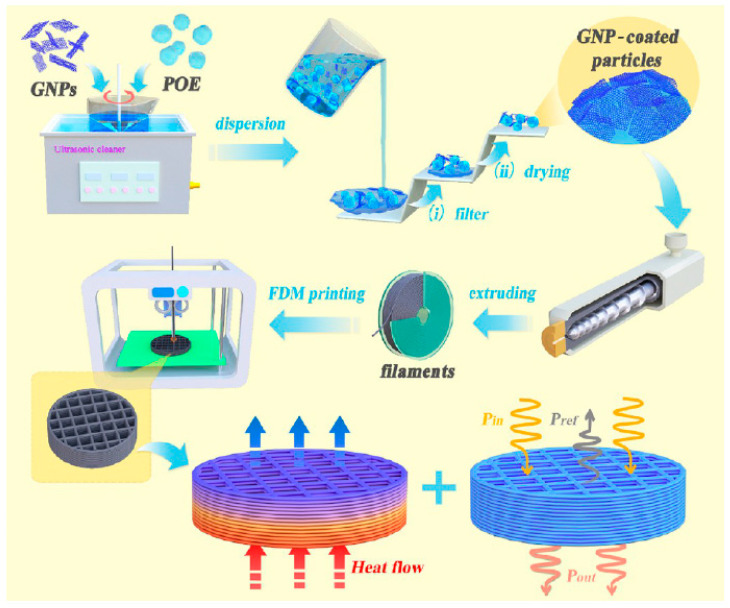
Scheme showing the preparation of POE-GnPs filaments and the FDM 3D printing process [25]. With permission from American Chemical Society.

**Figure 15 polymers-16-00195-f015:**
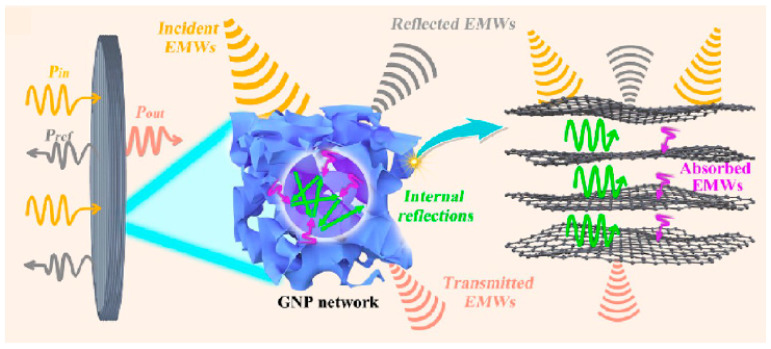
Scheme showing the absorption/multiple reflection EMI shielding mechanism of POE-GnP foams [25]. With permission from American Chemical Society.

**Figure 16 polymers-16-00195-f016:**
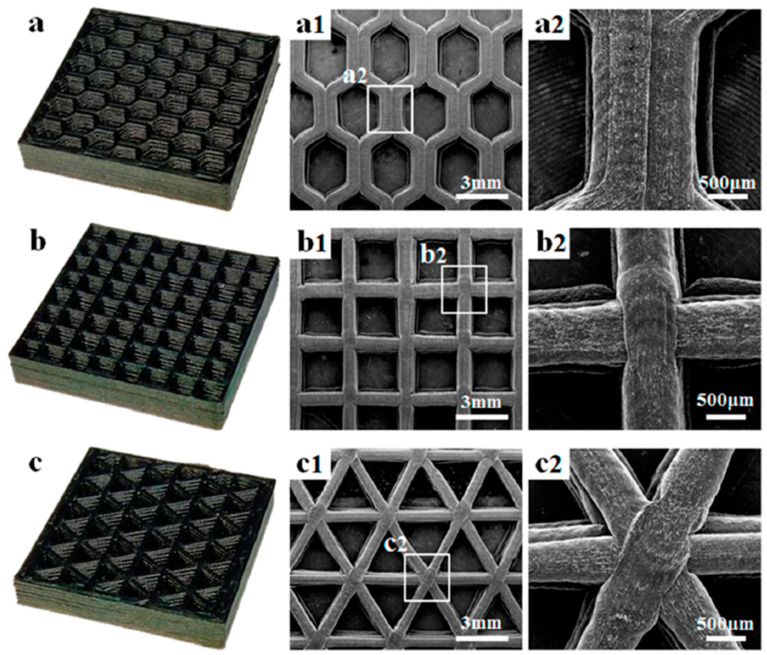
Digital images and SEM micrographs of FDM 3D-printed cellular honeycomb-like structures with different cell geometries: hexagon (**a**,**a1**,**a2**), square (**b**,**b1**,**b2**), and triangle (**c**,**c1**,**c2**) [26]. With permission from American Chemical Society.

**Figure 17 polymers-16-00195-f017:**
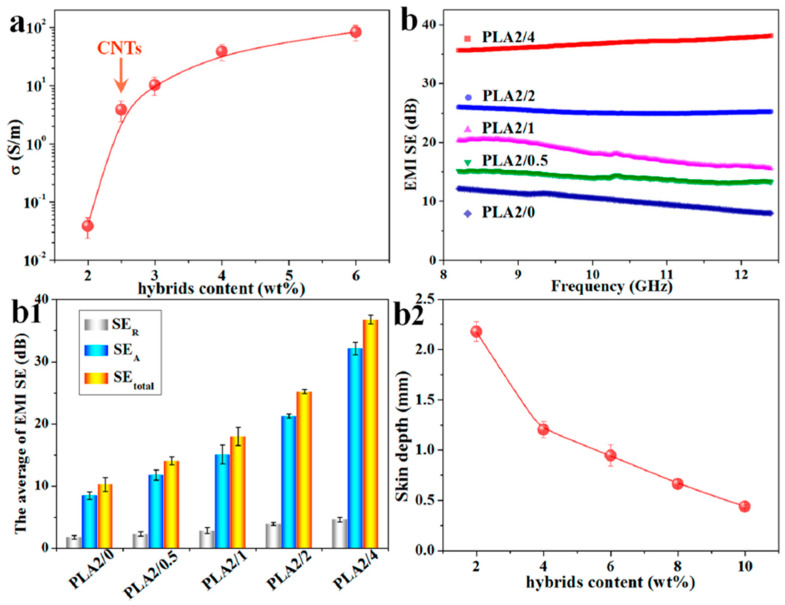
(**a**) Electrical conductivity and (**b**) EMI SE of PLA-GnPs-CNTs cellular nanocomposites; (**b1**) EMI shielding parameters including SE_total_, SE_R_, and SE_A_ and (**b2**) skin depth [26]. With permission from American Chemical Society.

**Figure 18 polymers-16-00195-f018:**
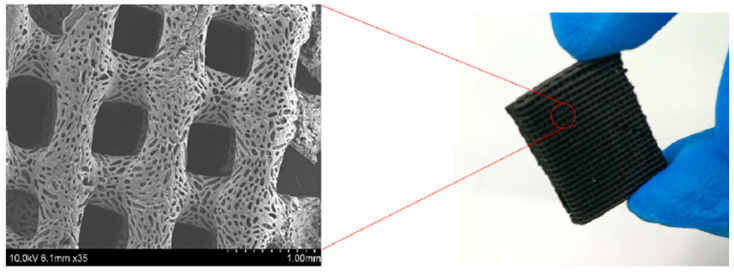
Three-dimensional printed CS-CNTs: SEM micrograph (**left**) and mesh (**right**) [27]. Copyright 2023, Elsevier.

**Figure 19 polymers-16-00195-f019:**
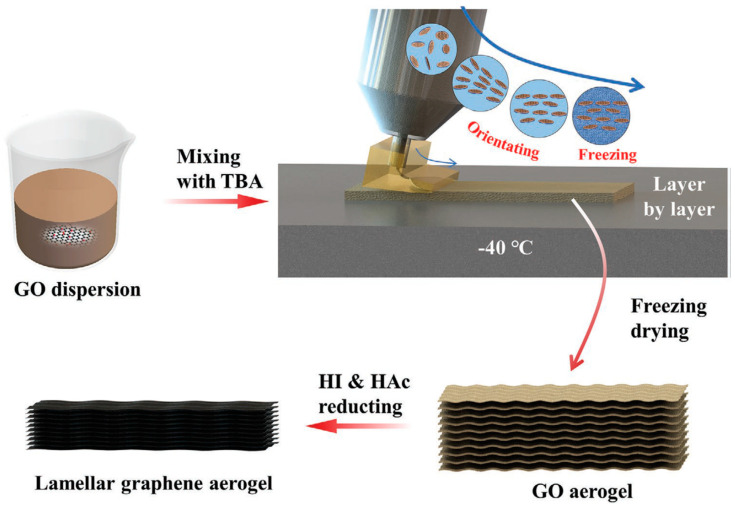
Scheme of the 3D printing LGA process, from the preparation of the GO dispersion to the final lamellar graphene aerogel [28]. Copyright 2023, John Wiley & Sons.

**Table 1 polymers-16-00195-t001:** Recently investigated polymeric foams and porous structures for EMI shielding.

Polymer	Nanofiller(s)	Foaming Method	EMI SE	Key Innovation	Ref
PVDF	CNT	Solid state scCO_2_	30 dB·cm^3^/g	- Reflection to multiple reflection/absorption mechanism - Lower percolation threshold	[7]
PS	CB, CNT, GnP	Microwave scCO_2_	>50 dB·cm^3^/g	- Absorption EMI shielding	[8]
PMMA	GO-NiNCs	Solid state scCO_2_	53 dB	- Absorption EMI shielding	[9]
rPET ^1^	SWCNT	Solid state scCO_2_	210 dB·cm^3^/g	- Great durability - Recyclable/re-processable/re-foamable	[10]
PA6	CF	Chemical injection molding	37 dB	- EMI SE 30% higher than non-foamed material - Absorption-dominated mechanism	[11]
PP	CNS	Core back injection molding		- Enhanced specific bending modulus	[12]
PP	MXene/rGO	Solid state scCO_2_	>25 dB	- Almost full absorption mechanism	[13]
PMMA	GnP/CNT	Solid state scCO_2_	>35 dB	- Almost full absorption mechanism	[14]
PU	CNT/Ni particles	Dip-coating	>42 dB	- Higher EMI SE when compared with foams comprising only CNTs	[15]
PU	Fe_3_O_4_-PVA/GO-Ag particles	Dip-coating	>30 dB	- Almost 280 dB·cm^3^/g	[16]
PU	SWCNT	Double dip-coating	56 dB	- Shape memory foam with modulated EMI SE - High durability - Adjustable EMI SE (18–30 dB)	[17]
TPU	CNT	Solution mixing, layer-by-layer casting, hot pressing	>35 dB	- Multilayer gradient structure - Higher EMI SE than homogeneous TPU-CNT	[18]
PCL-PLA	CNT	NaCl–water washing	23 dB (90 dB·cm^3^/g)	- Biodegradable foams - Selective distribution in PCL	[19]
PEI	CNT	Solid state scCO_2_	>30 dB	- Extremely low percolation threshold (0.06 vol%)	[20]
PP	CNT	Core back injection molding	60 dB	- Higher EMI SE due to CNT alignment	[21]
Styrene, BA, DVB and EHMA	rGO	Emulsion template		- Extremely low percolation threshold (0.055 vol%)	[22]
PDMS	CNT-Ni	Vacuum-assisted potting	45 dB	- Formation of biconductive network - High EMI shielding durability	[23]
PLA	GnP/CNT/CB	FDM 3D printing	35 dB	- Higher EM absorption capability	[24]
POE	GnP	FDM 3D printing	35 dB	- Enhanced EM absorption - High thermal conductivity	[25]
PLA	GnP/CNT	FDM 3D printing	37 dB	- Improved mechanical performance	[26]
CS	CNT	FDM 3D printing	>25 dB	- Absorption loss > 19 dB	[27]
Aerogel	GO	FDM 3D printing	69 dB	- Possible use for sensor applications	[28]
PI aerogel	MXene-CNT	FDM 3D printing	>68 dB	- Almost full absorption mechanism	[29]

^1^ rPET—Chemically recycled PET.

## Data Availability

Not applicable.

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
