# Peer review of "Recent Trends in Polymeric Foams and Porous Structures for Electromagnetic Interference Shielding Applications"

_polymers, 2024, doi:10.3390/polym16020195_

Round 1

Reviewer 1 Report

Comments and Suggestions for Authors

The manuscript was well written and presented in a easy way. However, the usage of AI and ML in polymeric nanocomposities design needs to be added as separate list in future presecpective. This would increase the reability of the mansucript. 

Comments on the Quality of English Language

Language is good. 

Reviewer 2 Report

Comments and Suggestions for Authors

Report
polymers-2781821

Recent trends on polymeric foams and porous structures for EMI shielding applications  
by
Marcelo Antunes

The author reviews the application of polymer based foams to protect electronic devices against electromagnetic interference (EMI).
This work  intends to review the foam containing conductive ingredients and find out a connection between foam microstructures and properties such as electrical conductivity/shielding efficiency. The author also considers polymer foams as templates for carbon foams for high-temperature EMI shielding applications. How does carbon foam withstand high temperatures? What temperature is meant?

All images are from other articles; There are no illustrations created by the author himself.
Please make sure that all images from the sources cited have been approved in writing by the editorial team for inclusion in the article!
As a reviewer, I do not have access to review this issue.

The abstract is long, complicated and not well structured. It should succinctly state the period of literature search for this review, the scope, the main discussed branches and sub-branches.

Title
Abbreviations such as EMI (electromagnetic interference) should be avoided in the title.
Ppolymeric foams have porous structures, there is no necessary to include both "polymeric foams and porous structures" in the title.

A review should not only summarize existing work, but also provide perspective and relevant discussions on the topic.
The reported work that show significant steps in topic development should be included in a table.

Reviewer 3 Report

Comments and Suggestions for Authors

In this review, the authors summarized the recent research progressing on polymer composite-based foams for EMI shielding applications. This work systematically reviewed the recent development in this topic in terms of foaming technology, filler innovation, computation-based structure optimization, and printing technology. The writing is very organized and easy to follow. The attenuation and thermal conducting performance of certain works were clearly listed to address the results from literature work. Therefore, I am happy to recommend this article being accepted in the present form. One suggestion is that, a table with key performance factor, e.g., EMI shielding efficiency, thermal conductivity, key innovation point, and literature reference number is suggested to be added to present a clear take-away to the readers. But it's optional.

Reviewer 4 Report

Comments and Suggestions for Authors

Polymer-based (nano)composite foams, especially those with a microcellular structure with suitable conductive (nano) fillers, have been shown to be excellent shielding materials and therefore reduce electromagnetic interference (EMI). It is promising as a shield for electronic devices to limit/avoid contamination.However, due to the high (micro)structural complexity associated with their multiphase nature, there is still much ignorance regarding the shielding mechanisms that operate in these materials, and therefore between the relationship of microstructure, cellular/porous structure and properties.It is necessary to study the relationship between (EMI SE), especially in terms of conductivity and EMI shielding efficiency. The paper is well written and rigorous and addresses the problems significantly.
